# Help or Hindrance? The Alcohol Industry and Alcohol Control in Portugal

**DOI:** 10.3390/ijerph16224554

**Published:** 2019-11-18

**Authors:** Maria Margarida Paixão, Mélissa Mialon

**Affiliations:** 1Public Health Unit of Amadora, ACES Amadora, Regional Health Administration of Lisbon and Tagus Valley, Praça Conde da Lousã, 2720-120 Amadora, Portugal; 2Faculty of Public Health, University of São Paulo, Av. Dr. Arnaldo, 715 – Cerqueira César, São Paulo – SP 01246-904, Brazil; melissa_mialon@hotmail.fr

**Keywords:** policy dystopia, alcohol industry, corporate political activity, alcohol control policy

## Abstract

The influence of the alcohol industry, also known as “corporate political activity” (CPA), is documented as one of the main barriers in implementing effective alcohol control policies. In Portugal, despite an alcohol consumption above the European average, alcohol control does not feature in the current National Health Plan. The present research aimed to identify and describe the CPA of the alcohol industry in Portugal. Publicly-available data published between January 2018 and April 2019 was extracted from the main websites and social media accounts of alcohol industry trade associations, charities funded by the industry, government, and media. A “Policy Dystopia” framework, used to describe the CPA strategies of the tobacco industry, was adapted and used to perform a qualitative thematic analysis. Both instrumental and discursive strategies were found. The industry works in partnership with health authorities, belonging to the national task force responsible for planning alcohol control policies. Additionally, it emphasizes the role alcohol plays in Portuguese culture as a way to disregard evidence on control policies from other countries. This paper presents the first description of CPA by the alcohol industry in Portugal and provides evidence for the adoption of stricter control policies in the country.

## 1. Introduction

Portugal has an alcohol consumption above the average for the World Health Organization (WHO) European region [1] and wine represents more than half of all alcohol consumed [1,2,3]. In 2016, 3486 deaths were attributed to alcohol abuse in the country and there was a prevalence of 6.8% of alcohol use disorders [3]. In 2012, the Portuguese General-Directorate of Health developed a National Health plan, which was revised in 2015, establishing the priorities and goals for public health efforts in the country until 2020 [4]. In this document there is no mention of alcohol control, contrary to an official recommendation by the General-Directorate for Intervention on Addictive Behaviours and Dependencies [4,5]. In 2018, there was a proposal to increase the taxes on alcoholic beverages in the country (excluding wine), but the proposal was highly contested by the alcohol industry and was eventually called off before the final presentation of the measures implemented in the Portuguese state budget for 2019 [6].

It has been documented that one of the main barriers to the development of effective alcohol control policies is the influence of corporations in the alcohol industry [7,8]. The concept of “corporate political activity” (CPA), used to describe the influence of the tobacco, food, and gambling industries, among others, on public health policy, research, and practice, has been explored in the literature [9,10,11,12,13]. A framework was developed by Ulucanlar et al. to describe the CPA of the tobacco industry [12]. The same framework was later used and revised to analyse the influence of the food industry [9]. It provides a classification that helps understanding and categorising strategies used by industry actors to create policy dystopia and is divided into “instrumental” (action-based) and “discursive” (argument-based) strategies [12]. 

For the alcohol industry, there is evidence of the use of similar strategies to influence policy, research, and practice [11,14]. Also described in the literature is the creation and dissemination of misleading information related to health risks associated with alcohol consumption, especially the increased risk of cancer [15,16], and a focus on the health benefits of alcohol intake [11,15,16]. Hessari et al. provide evidence of a shift away from the harm associated with alcohol drinking for women, in particular breast cancer [16]. Consequently, understanding and documenting the CPA of the alcohol industry is a keystone to developing alcohol control policies to reduce the impact of alcohol on the population’s health [17].

To date, the majority of the evidence published regarding the influence of industries on public health policy, research, and practice comes from English-speaking countries, including the United Kingdom, the USA, Australia, and New Zealand, as highlighted in a recent systematic review [14]. In these countries, beer is the most consumed alcoholic drink [18]. In the case of Portugal, alcohol control has been described as low, with wine being both the most consumed alcoholic drink and an important sector of the country’s economy [19,20]. Additionally, there is evidence that the interaction between lobbyists from another sector, the tobacco industry, and policy-makers led to delays in the development and implementation of public health policies in the country [21]. Hence, there is a need and space for research originating from different cultural contexts, namely southern European countries with cultural traditions associated with wine production and consumption. The use of an existing framework developed by Ulucanlar et al. [12] could also help to infer about generalisability. 

In this study, we aimed to identify the current CPA of the alcohol industry in Portugal.

## 2. Materials and Methods 

Data were collected between May and June 2019. To be included in the study, data had to be published between January 2018 and April 2019. The decision to exclude earlier data was made because earlier publications may not depict current practices. 

The methodology developed by Mialon et al. [22] to identify the CPA of the food industry was used in this study. The proposed data collection and analysis comprised five steps: (1) selection of a sample of industry actors, (2) identification of sources of information, (3) data collection, (4) data analysis, (5) dissemination of findings and identification of possible policy implications.

For this study, we selected trade associations from different sectors of the alcohol industry (wine, beer, and liquors) on the basis of the number of producers they represented and their importance in the Portuguese market. Only trade associations were considered in order to depict practices by large organisations in the country. The following trade associations were selected: Associação de vinhos e espirituosas de Portugal—ACIBEV (Wine and Spirits Association of Portugal), Organização Interprofissional do Vinho de Portugal (Interprofessional Organisation of Wine in Portugal), Sociedade Central de Cervejas (Central Society of Beers), Associação dos Cervejeiros de Portugal (Brewers’ Association of Portugal), Associação Nacional de Bebidas Espirituosas—ANEBE (National Association of Spirit Drinks). The alcohol industry social aspects/public organisations (SAPRO) “Beba com cabeça” and “100% Cool” were also included. The justification for the inclusion of each actor is detailed in the Appendix A. 

The list of sources used for this study is available in Appendix A. We collected information on each alcohol industry actors’ websites and social media accounts. We also included information from the websites of the Ministry of Health, the Portuguese parliament, and the General-Directorate for Intervention on Addictive Behaviours and Dependencies. The websites of universities in the country that taught medicine and/or public health were also included. The five highest ranked newspapers in the country were selected as well [23]. In these websites, the search strategy used a combination of the word alcohol and/or the names of the industry actors selected for the study ((álcool) OR ((álcool) AND (Associação de vinhos e espirituosas de Portugal OR ACIBEV OR Organização Interprofissional do Vinho de Portugal OR Sociedade Central de Cervejas OR Associação dos Cervejeiros de Portugal OR Associação Nacional de Bebidas Espirituosas OR ANEBE))).

A qualitative thematic analysis was conducted, using Microsoft Excel to manage the data (version 2016, Microsoft, Redmond, Washington, USA). We used a deductive approach to data analysis, and the framework is presented in Appendix A. This framework was adapted from the model of Ulucanlar et al. [12] for the tobacco industry and the frames and arguments used by the alcohol industry identified by Savell et al. [11]. Additionally, we included information from another framework developed by Savell et al. [11] for classifying the CPA of the tobacco industry, which has been used for the food industry by Mialon et al. [9] and for the gambling industry by Hancock et al. [24] 

Data collection and analysis was led by the lead author, a Portuguese native speaker, with initial guidance from the second author. Subsequently, the second author, in each of the two rounds of revisions, revised a random sample of 10% of the data. There was disagreement about the coding for two records. This was resolved after discussion between the authors. Data extraction tables were adapted from the methods developed by Mialon et al. to identify the CPA of the food industry [25]. All data collected is available as Appendix A. In this manuscript, we use references starting with the letter A when presenting data from Appendix A. We offer here a narrative presentation of the main results and examples collected to illustrate our findings. In the text, quotations were translated from Portuguese to English by the first author and revised by the second author. 

## 3. Results

In total, 198 written records were collected and analysed for this study. Websites and social media pages of trade associations represented most of the information identified for the CPA of the alcohol industry in Portugal (approximately 68%), followed by SAPRO (17%). Government material and media material represented 13% and 3%, respectively. No information was identified on universities’ websites. Overall, we found both instrumental (56%) and discursive (44%) strategies. 

### 3.1. Instrumental Strategies

The instrumental strategies identified are listed in Table 1.

Alcohol industry actors interacted with entities that otherwise could have interests opposed to those of the industry, including health organisations. Representatives of the industry founded and continued to be part of the national forum on alcohol and health. They hold monthly meetings with the national health authorities to discuss public policy (**A1, 133**). Information published on both the industry’s and the national forum’s websites state that the industry shares the same objectives as health authorities (**A1, A132**). At least two trade associations are also part of the executive commission of this forum (**A1, 95**), having a leading role in its activities: 

“ANEBE, as a founding organisation and current member of the executive commission of the National Alcohol and Health Forum, shares the same goals as the health community” (**A1**).

“Cervejeiros de Portugal (…) is a member of the executive commission of the National Alcohol and Health Forum and a member of the National Council for the Coordination of drugs, addiction and harmful use of alcohol” (**A95**).

We identified references to meetings and initiatives with representatives of the General Directorate for Intervention on Addictive Behaviours and Dependencies (**A30, 31, 58**) with a regional secretary of health (**A33**) and with secretaries of state (**184**) to emphasize the importance of an “intelligent alcohol consumption” (**A30**). ACIBEV also mentioned on their website a meeting with the Secretary of State for the European Affairs with whom European policy on alcohol was discussed (**A82**).

There is a record on both ACIBEV’s and the Parliament’s website inviting the President of the Parliamentary Commission on Agriculture and Sea (**A163**), the Secretary of State for Civil Protection and the Assistant of the Secretary of State for Mobility (**A184**), the sub-director for the General Directorate for Intervention on Addictive Behaviours and Dependencies (**A57**), and the General-Inspector of the National Authority for Food and Economic Safety (**A57**) to a “sunset with wine” to “celebrate the longer and warmer days of summer” (**A57**).

There is a record of a meeting in January 2018 with the Portuguese ambassador for the Organisation for Economic Co-operation and Development (OECD) to “discuss the concerns of the sector…” (**A85**) and Cervejeiros de Portugal refers to a “privileged contact” with the Ministry of Health on their website (**A95**). Also, a SAPRO, 100% Cool, refers to an informal meeting with the President of the Portuguese Republic at a public event in February 2018 on their social media page (**A195**). 

Industry representatives organised scientific conferences partnering with the Portuguese Association of Nutrition (**A113**), inviting researchers and stating their affiliations in order to establish credibility (**114**). Similarly, when contacting the media about the health impacts of alcohol, the Associação dos Cervejeiros de Portugal transmitted messages via an expert, identified as a nutritionist, with the same article referring to her additional work as a consultant for the industry on health issues (**A142**).

We found evidence that the alcohol industry collaborates with the Portuguese police and security forces. This was observed extensively, especially through the SAPRO “100% Cool” which had routine operations with security services, awarding drivers who had no alcohol in their breath and giving out information on moderate and responsible alcohol consumption (**A26, 27, 50, 141, 164-172, 175, 179-80, 182, 186, 189, 192-3**). 

There were several examples of involvement of the alcohol industry in the community through initiatives in different areas such as cinema (**A91**), cultural events (**A48**), and music events (**A190**). 100% Cool was present in the major universities’ festivities in the country (**A170, 173, 174, 176**) and Associação dos Cervejeiros de Portugal refers to a partnership with the national parents’ association, the national teachers’ association, and with the Portuguese Institute of Youth (**A96**). It is also a founding member of the Portuguese society for recycling (Sociedade Ponto Verde) (**A94**).

The ANEBE ordered a market study from a consultancy firm, which was delivered to the Portuguese Parliament upon contestation of a possible increase in taxes on alcoholic beverages. This concluded that increases in taxes would not be beneficial and would even harm the national economy (**A39, 148, 162**). The study was extensively used by the alcohol industry to lobby against the proposed changes in the taxation. To our knowledge, no other study was taken into consideration by the Portuguese Parliament. Eventually, the measure was not adopted by the Government. (“Taxes were frozen according to rein vindications by beer and liquor drinks producers (…) according to the proposal for the State Budget of 2019, the Government heard their arguments” (**A198**)).

### 3.2. Discursive Strategies

Discursive strategies are depicted in Table 2. Other arguments which did not fit in the existing CPA/policy dystopia classification but were relevant to the analysis are listed in Table 3. 

The comparison to neighbouring Spain’s tax policy on alcoholic beverages was found repeatedly, both to emphasize the unfairness of any increase in taxes and to argue for their decrease to stop illicit trade (**A28, 122, 159**). This was found both in the websites of two trade associations **(A28, 122)** and in the study paid for by ANEBE delivered to the Portuguese Parliament (as previously described) (“… aligning the tax in a community perspective, regarding Spain, will allow a reduction in smuggling and cross-border shopping” (**A159**)). 

The importance of self-regulation by the industry was frequently mentioned as a necessity and that it was justifiable given the claim of the industry as a responsible one (**A16, 49, 59, 70, 90, 107**). Self-regulation was also cited by the ACIBEV as a crucial part of creating value in the production of wine (**A90**). In Portugal, it appeared that all these trade associations were self-regulating their activities.

There were several mentions of the benefits of alcohol, without an adequate portrayal of the harms posed by it. Indeed, we found references to alcohol being an important component of a healthy diet (**A101, 103, 126**) and good for one’s health (“…the consumption of beer is beneficial to one’s health…” (**A115**)). 

Beer was portrayed as an important source of vitamins and water for the body, including being mentioned as a way to satisfy thirst (**A105**). Wine and beer components were mentioned in the industry’s websites as protective factors in cardiovascular diseases (**A87**), Parkinson’s (**A87**), Alzheimer’s (**A87**), diabetes mellitus type 2 (**A88, 97**), nephrolithiasis (**A97**), eye cataracts (**A97**), anaemia (**A124**), osteoporosis (**A124**), and as a way to boost the immune system (**A124**). 

One record was found mentioning the association between alcohol consumption and breast cancer (**A196**). However, the association was misrepresentative, avoiding the scientific and the common nomenclature for breast cancer in Portuguese, referring to it as “chest cancer” (**A196**).

The concept of a risky consumption was widely used to justify the need for an individual-level approach (**A71, 73, 79, 80, 86**) to minimise the health impacts of alcohol (**A134**) and to make a connection between the cultural role of alcohol in Portugal in comparison to other countries (**A70**). As such, any population-level approach was branded as unfair to responsible drinkers (**A72**). 

Another practice we found was the creation of an online calculator of the concentration of blood alcohol by ANEBE (**A9**). This tool was marketed as a way for consumers to be responsible and aware of their alcoholic blood concentration before driving, putting an emphasis on individual behaviour and responsibility (**A9**). A topic repeatedly found was the cultural comparison between Portugal and other countries, with the argument that in Portugal, similarly to other southern European countries and in contrast to northern ones, alcohol is part of the history and culture, hence the consumption is moderate, harmless, and part of the lifestyle (**A16, 52, 68, 138**).

“The consumption of alcohol varies significantly between European countries. In Southern Europe (Cyprus, Greece, Italy, Malta, Spain and Portugal) there is a wine-making tradition, with daily consumptions that accompany meals and in general, a repudiation of drunkenness in public” (**A138**).

This was also used to justify why stricter public measures regarding alcohol control probably did not need to be applied in Portugal and in countries with similar traditions: 

“Southern countries must unite, because when measures are applied there will not be a distinction between wine and vodka.” (**A79**).

Through interviews in the media, industry’s representatives voiced the danger of Portugal following the example of other countries, citing for example Ireland’s recent policy that introduced health warnings on alcohol beverages’ labels (**A43, 53, 55, 56, 84**). Industry actors framed the policy as a war on the industry “anti-alcohol lobbies want to deny us a glass of wine with the meal.” (**A75**). They also considered it inadequate in the Portuguese context, inclusively being framed as a cultural imposition:

“we alert for the danger of increasing the price and restricting advertising on alcoholic drinks, as has happened in northern European countries” (**A55**).

“there is a web of activity, involving several entities who want to create a restrictive environment on alcohol and impose a northern culture on southern European countries.” (**A65**).

## 4. Discussion

The results of this research show a close cooperation between health authorities and representatives of the alcohol industry, with the creation and regular meetings of the National Health Forum of Alcohol and Health. This task force is responsible for planning alcohol control policies in the country. Industry members being part of its executive commission has probably led to interference with public health measures. It also provides legitimacy to the industry. The close cooperation between health authorities and the industry is similar to what had been described for the tobacco industry in Portugal [21,26]. Ravara et al. called the process of discussion between stakeholders for the adaptation of tobacco control legislation as “being far from transparent” [21].

The use of the argument that moderate consumption can be harmless or even beneficial to health was extensively found. This has controversial evidence [27]. A study estimated that the burden of alcohol was larger than previously expected, on the basis of data from the Global Burden of Disease project from 195 countries [28]. Indeed, the small protective effects alcohol might have for cardiovascular disease risk are largely outweighed by the overall negative impact it has on broader population health [29]. This evidence goes against the thresholds of risk defended by the alcohol industry and against the notion of patterns of consumption being key for public health interventions. Consequently, in future public health interventions the key messages for the population could shift from a moderate consumption to the importance of restricting alcohol to the minimum possible, given there is no safe limit [29].

This research also finds similar results to other papers that depicted an avoidance of the reference of the association between breast cancer and alcohol consumption [16]. In the case of this paper, the association was also concealed with the use of misleading language, which was described in other contexts as well [30]. Therefore, the exploration in health education efforts of this association is necessary.

The finding of inaccurate depictions of health risks and benefits also raises the question of how effective self-regulation is by the industry regarding their online presence. It appears that there is no audit of the industry’s websites or social media pages. As such, health authorities in Portugal could audit the information posted online by actors affiliated to the alcohol industry as they could have an impact on population’s health literacy and undermine public health efforts.

We also urge authorities to investigate the industry’s claims of an increase in illegal production and smuggling of products from neighbouring Spain and other countries. This statement by the industry was not found to be backed by any research.

This research highlights the use of cultural arguments as a defence mechanism by the alcohol industry. Indeed, it presents an easy way to connect with the public and provides a sense of legitimacy to the industry. The cultural argument makes public health efforts more difficult, given the fact that these need to be careful not to fall into the “authoritarian” speech of wanting to control the population’s behaviour and preferences, disregarding their cultural heritage, as claimed by the industry [12]. 

Public health initiatives would also face the additional challenge of recognising the importance alcohol production (namely wine) has in a lot of rural areas of the country and how much it is part of the cultural identity of its people. We urge national authorities to develop interventions aimed at the deconstruction of the cultural arguments referred by the industry [31,32]. The cultural claim is intrinsically bridged with the argument about patterns of consumption. According to literature, despite the country’s high consumption of alcohol, it has a lower prevalence of binge drinking compared to the European average [19], with 11% prevalence of binge drinking amongst the population in 2015. This was frequently used by the industry as a reason why Portugal did not need a population-level approach to alcohol control.

Despite not having found absolute evidence of influence on national policy, the delivery of a study, paid for by the industry, to the Portuguese Parliament around the time of a discussion on a possible increase in taxation of alcoholic beverages could suggest that this effort was effective, as claimed by national media [6]. 

In England, a partnership between Public Health England with a SAPRO led to criticism that it could undermine possible public health interventions and that it legitimised an industry whose main objective is profit [33,34]. However, in Portugal, as far as we are aware, no such criticism was made of the official collaboration with the alcohol industry. Consequently, this paper hopes to highlight the need to think critically about the opportunities and risks such partnership has offered so far in Portugal and what can be learned from other countries.

However, it can be difficult to exclude the industry from social initiatives given the fact that government authorities often lack the financing for such initiatives. This raises the question of how much social responsibility should be left to industry actors [35,36]. In the case of Portugal, after overcoming an economic crisis with cuts to public health interventions, it could be argued that industry actors financing social initiatives could provide the only source for some public health interventions. As such, the health authorities of the country need to start debating which areas affecting public health are being neglected by state funding, giving the space to private initiatives to take the lead and replace official authorities. Another possible solution for chronic underfunding of public health efforts could be earmarking resources from industry’s taxes.

To our knowledge, this was the first study to identify strategies employed by the Portuguese alcohol industry to influence alcohol control. The main limitation of this research is that it is based on publicly-available information that can depict a different reality from internal documents. This could especially be the case for government and universities where we found little public information. Additionally, there could be some potential for biases given that not all the material coded was reviewed by both authors.

## 5. Conclusions

Our findings suggest that the low control policies regarding alcohol in Portugal are probably associated with industry interference, through a preference for self-regulation, an emphasis on education, and the production of evidence. The fact that so many of the tactics, such as meetings with health organisations and policymakers and a misrepresentation of health effects of alcohol, were available online makes it obvious that the industry acts with relative impunity. As such, the use of the evidence described in this article could help build more effective alcohol control policies in the country and counteract the public arguments used by industry’s representatives.

## Figures and Tables

**Table 1 ijerph-16-04554-t001:** Instrumental strategies of the alcohol industry in Portugal.

Argument Identified	Number of Occurrences	Reference in the Manuscript
Amplification of information	28	A23, 29, 38, 39, 87, 88, 97, 98, 99, 100, 101, 103, 104, 105, 112, 115, 116, 117, 118, 123, 124, 126, 127, 144, 145, 153, 159, 160
Community involvement	44	A6, 7, 11, 12, 26, 27, 42, 48, 50, 91, 96, 110, 111, 140, 141, 164, 165, 166, 167, 168, 169, 170, 171, 172,173, 174, 175, 176, 177, 178, 179, 180, 181, 182, 185, 186, 187, 188, 189, 190, 191, 192, 193, 194
Credibility of information	6	A114, 128, 142, 143, 146, 162
Internal constituency recruitment	7	A3, 13, 21, 46, 77, 84, 93
Production of information	1	A37
Relationship with leaders/health organizations	21	A1, 30, 31, 33, 40, 44, 45, 57, 78, 82, 83, 85, 94, 95, 113, 132, 133, 139, 163, 184, 195
Relationship with media	1	A183

**Table 2 ijerph-16-04554-t002:** Discursive strategies used by the alcohol industry in Portugal.

Argument Identified	Number of Occurrences	Reference in the Manuscript
Discriminatory among the producers	4	A18, 36, 108, 120
Existing regulation is satisfactory	1	A10
Government is unreasonable	2	A119, 155
Illicit trade	5	A28, 122, 149, 157, 161
Importance of the industry to the economy	9	A14, 19, 25, 34, 60, 121, 148, 154, 156
Interferes with a free market economy	1	A35
Lost/unreliable tax revenue	7	A22, 24, 147, 150, 151, 158, 198
Not enough evidence	1	A67
Policy will not work	1	A152
Self-regulation/education	30	A2, 4, 5, 8, 9, 15, 20, 32, 41, 47, 49, 51, 58, 59, 69, 79, 80, 81, 86, 89, 90, 92, 102, 106, 107, 109, 125, 129, 130, 131
Unfair to drinkers	1	A72

**Table 3 ijerph-16-04554-t003:** Discursive strategies identified that did not fit into the previously constructed framework.

Argument Identified	Number of Occurrences	Reference in the Manuscript
Other/culture	10	A16, 52, 61, 62, 65, 68, 70, 72, 74, 75, 76
Other/pattern of consumption	7	A17, 66, 71, 73, 134, 135, 136, 137, 138, 196
Other/suppression of information	1	A64
Other/war on alcohol	10	A43, 53, 54, 55, 56, 63

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
