# Peer review of "Help or Hindrance? The Alcohol Industry and Alcohol Control in Portugal"

_ijerph, 2019, doi:10.3390/ijerph16224554_

Round 1
Reviewer 1 Report
The article entitled: “Help or Hindrance? The alcohol industry and alcohol control in Portugal” concerns the socially important issue of political activity of the alcohol industry enterprises.
The article was correctly formatted, divided into parts in accordance with the requirements of the journal. It is written in an interesting way and has appropriately selected, widely analyzed and current literature. It is an important scientific point of view of activities of the alcohol industry in Portugal and is a social voice of reflection regarding its goals and scope of control.
The work shows an excellent ability to deal with the complexity of the presented topic, which is communicated to the reader in an accessible form.
It is worth supporting the diligence of the publication before publishing it by supplementing the content of the article with possible limitations of this work.
The text is worth publishing, which I recommend.
Author Response
Thank you for your comments and recommendation of our paper for publication. Point 1: "It is worth supporting the diligence of the publication before publishing it by supplementing the content of the article with possible limitations of this work." As suggested, we have expanded our limitations by including a comment on the coding approach used, specifying that there could be some potential for biases due to some of the material coded not having been reviewed by both authors. We hope you will consider our article ready for publication after these changes. Kind regards, The authorsReviewer 2 Report
In this manuscript the authors collected data from websites and social media accounts of alcohol industry trade associations in Portugal to investigate how alcohol industry influenced Portuguese health authorities to have no population-level policy control upon alcohol consumption in this country. Overall the authors have done a solid work. Other than a few small issues, I don’t think I need to review it again.
(1) There are some one-sentence paragraphs in the current manuscript. The authors should find ways to combine them with former or latter ones.
(2) A "%" is missing after "13" in line 113.
(3) The sentence in lines 121-123 has grammatical mistake and should be revised.
Author Response
Thank you for your comments and suggestions on how to improve the article. We combined several one-sentence paragraphs and the changes proposed in (2) and (3) were made. We hope you will consider our article ready for publication after these changes. Kind regards, The authorsReviewer 3 Report
Although I found the topic of utmost relevance, I think that the methods section of this article has serious flaws that should be amended prior to publication. Please find my specific comments below:
Introduction: The authors could improve this section to make clearer to the reader which is the current gap in the literature that justifies the need for this study. Line 47: Typo; it should read "divided into", not "divided in" Line 59: Which particularities of the non-English-speaking countries are the authors referring to? Some context would be useful here to understand why the results are not generalizable to Portugal. Please state the research objective more clear, and avoid using references when specifying the overall aim. Line 66: Neither the choice of the study design nor the theoretical perspective are identified or stated in the Methods section Line 72: be consistent with the use of past terms (analysis comprised five steps, not is comprised of five steps). Line 90-92: Please provide the search strategy Line 93: Authors should provide the necessary details about how the analysis reduced the data into workable themes and the emerging conclusions. Avoid using references in your conclusions.Author Response
Thank you for your comments and suggestions on how to improve the article. We have modified the introduction in order to better justify the relevance of the paper. We changed the research objective in order to make it clearer and without references. We corrected the typographical error identified in line 47. Also, we added in the end of this section that the present study is a qualitative thematic analysis. Our methods section was expanded to include details that hopefully will help to clarify how the analysis was conducted and how it led to our conclusions. Also the search strategy used was included. The verb tense in line 72 was corrected. Our conclusions were reformulated in order to better summarize the importance and implications of the study and the references were removed from this section. We hope you will recommend our article for publication after these changes. Kind regards, The authorsRound 2
Reviewer 3 Report
The authors have addressed all of my previous comments and suggestions. Thus, I do believe that this manuscript is suitable for publication.